# The Burden of Sleep/Wake Disorders: Excessive Daytime Sleepiness and Insomnia Project

**DOI:** 10.3390/mps7050070

**Published:** 2024-09-06

**Authors:** Marina Tüzün, Ulf Kallweit, Stefan Seidel, Olga Endrich, Sven Trelle, Maurizio A. Leone, Oliviero Bruni, Richard Dodel, Maria Konti, Maria Lolich, Elisabetta Pupillo, Dauren Ramankulov, Luca Vignatelli, Carla Meyer-Massetti, Markus Schmidt, Claudio L. A. Bassetti

**Affiliations:** 1Interdisciplinary Sleep-Wake-Epilepsy-Center, Bern University Hospital (Inselspital), 3010 Bern, Switzerland; marina.tuezuen@insel.ch (M.T.);; 2Faculty of Medicine, University Witten/Herdecke, 58455 Witten, Germany; 3Center for Biomedical Education and Research (ZBAF), 58455 Witten, Germany; 4Rehabilitation Clinic Pirawarth, 2222 Bad Pirawarth, Austria; 5Medical faculty, Institute of Clinical Chemistry, 3010 Bern, Switzerland; 6Department for BioMedical Research, University of Bern, 3008 Bern, Switzerland; 7Medical Directorate, Clinical Trials Unit, University of Bern, 3012 Bern, Switzerland; 8Department of Neurosciences, Istituto di Ricerche Farmacologiche “Mario Negri” IRCCS, 20156 Milano, Italy; 9Department of Developmental and Social Psychology, Sapienza University, 00185 Rome, Italy; 10Department of Geriatric Medicine, University Duisburg-Essen, 45141 Essen, Germany; 11European Academy of Neurology, 1070 Vienna, Austria; 12IRCCS Istituto delle Scienze Neurologiche di Bologna, 40139 Bologna, Italy; 13Clinical Pharmacology & Toxicology, Department of General Internal Medicine, University Hospital Bern (Inselspital), 3010 Bern, Switzerland; 14Department of Neurology, Bern University Hospital (Inselspital) and University of Bern, 3010 Bern, Switzerland

**Keywords:** excessive daytime sleepiness, insomnia, sleep/wake disorders, quality of life, socio-economic burden

## Abstract

Excessive daytime sleepiness (EDS) and insomnia (IN) complaints represent the most common sleep/wake disorders. Currently, the specific needs of these patients and their relatives, as well as the overall socio-economic burden of IN and EDS remains widely unexplored. This pilot study to be carried out in Switzerland is a retro- and prospective, national, one-center cohort observational study for the systematic evaluation of the burden of EDS and IN and its evolution 12 months after the first assessment. Patient recruitment will be organized through 7–8 primary care providers (primary/general care practitioners and pharmacies). Primary outcomes are the prevalence of EDS/IN in the primary care setting and the association between EDS/IN with health-related quality of life (QOL) as assessed with the established instruments. Secondary outcomes are the association between EDS/IN with the presence of comorbidities, number of injuries/accidents, and number of sick/leave days for the subgroup of working subjects. Calculation of direct per-patient costs will be undertaken to analyze the economic implications of sleep/wake disorders, providing valuable insights into the financial burden experienced by affected individuals within the healthcare system. This research will provide information on the feasibility of such a study and inform on aspects of the QOL most associated with EDS/IN. Based on this pilot project, a European multicenter study on the burden of sleep/wake disorders will be conducted by the European Academy of Neurology.

## 1. Introduction

Sleep plays a crucial role in maintaining the health and well-being of individuals across all stages of life, from childhood through adolescence and adulthood [1]. Adequate sleep has been linked to improved cognitive function, mental health, and cardiovascular, cerebrovascular, and metabolic health [2]. Additionally, research has shown that sleep plays a crucial role in promoting neuroplasticity and bolstering immune functions [3,4]. Despite its importance, a large portion of the global population, over 36%, experiences sleep loss [5]. In Europe alone, it is estimated that 45 million individuals suffer from sleep disorders including sleep apnea affecting 3% of the population, chronic insomnia affecting up to 7%, narcolepsy at 0.022%, and hypersomnia at 0.75% [6]. The Swiss Health Interview Survey of 2012 also found that nearly a quarter of the Swiss population experiences difficulties with falling asleep, restless sleep, and frequent or early awakenings [7].

Sleep/wake disorders can have a significant effect on an individual’s physical and mental quality of life (QOL). The most significant factors associated with decreased QOL were high body mass index (BMI), cardiovascular disease, excessive sleepiness, initial insomnia, and use of sleep medication and antidepressants [8]. While some studies have shown significant improvement in QOL following continuous positive airway pressure (CPAP) therapy for patients with mild sleep apnea [9], other studies have not found a consistent overall difference in improvement between full and non-users of PAP [8].

Reimer et al. conducted a clinical review that revealed individuals with sleep disorders, particularly sleep apnea, narcolepsy, restless legs syndrome, and insomnia, generally experience a lower quality of life compared to the general population. Before treatment, their QOL scores tend to be comparable to those of individuals with chronic conditions such as hypertension and chronic obstructive pulmonary disease. The potential impact of treatment on QOL is uncertain and requires further investigation, as results may vary, with some individuals showing improvement to population norms while others do not [10].

Thus, although data on the economic burden of the major sleep disorders are available, there is a lack of data on the overall burden of the most frequent sleep/wake complaints, such as excessive sleepiness (EDS) and insomnia (IN) symptoms. And results of studies on the benefits of their diagnosis and treatment are controversial. The other problem is that there are not yet specific measures in common use for other sleep disorders.

It is also known that the economic burden of sleep disorders is comparable with other mental and brain disorders. One of the largest studies in Europe, which evaluated in 2010 the costs of disorders of the brain in Europe, showed that sleep disorders such as sleep apnea cost EUR 3860 per patient annually, and in the case of narcolepsy, the direct and indirect costs reach EUR 10,303 [6].

Taking into consideration the previously stated results, it should not come as a surprise that sleep/wake disorders represent an under-reported problem for both patients and doctors or healthcare providers. Studies indicate that sleep symptoms are exceedingly common among patients presenting for medical visits [11]. For example, the prevalence of insomnia among patients of general practitioners is about one-third higher than in the general population [12].

To summarize, while the socio-economic burden of specific sleep disorders is described, the overall burden of EDS and IN on the primary care level, along with the potential benefits of their diagnosis and treatment—such as improved quality of life, sleep quality, and daytime functioning—remains to be explored.

This pilot study aims to address the significant socio-economic and health burden of sleep/wake disorders with EDS and IN irrespectively of the underlying medical condition. The overall objective of the study is to evaluate the feasibility of recruiting patients with EDS and IN symptoms in the primary care setting and in cooperation with general practitioners and to evaluate if participants can be kept in the study for a duration of 12 months. The feasibility of collecting data during follow-up solely via online questionnaires will also be assessed. Based on the results of this pilot study, the European Academy of Neurology plans a larger European multicenter study on the burden of sleep/wake disorders.

## 2. Experimental Design

The protocol of the study was officially accepted by the Ethics Committee Canton Bern on 23 June 2023. This approval signifies that the study protocol has been reviewed and found to comply with ethical standards and regulations set forth by the Ethics Committee Canton Bern.

Study design and settings

The present pilot study is a retro- and prospective, national, single-center cohort observational study for systematic evaluation of the burden of EDS and IN in the primary care setting (approximate recruitment goal *n* = 100) and assessment of the changes after 12 months. During the study, participants will have the opportunity to receive therapy for their symptoms, although it is not the goal of the study to evaluate the effectiveness of the therapy. Based on this pilot study, a European multicenter study on the burden of sleep/wake disorders will be conducted by the European Academy of Neurology.

The current study’s screening and recruitment procedures were established and evaluated within the clinical routine at the center located in Bern.

Screening and recruitment of participants


*Prescreening and initial data collection*


We will recruit patients who show complaints of EDS or IN in the primary care setting and are interested in participating in the study. Because of the heavy workload and limited time available to consult with patients in physician offices, we will use a data collection sheet with a minimal number of questions that include just the presence of subjective sleep/wake complaints and which takes 1–2 min per patient for the primary care provider (PCP) to complete:

During consultancy in a PCP’s office, regardless of the primary reason for the visit, patients will be asked about the subjective presence of daytime sleepiness or insomnia symptoms in a sequential manner to avoid bias and estimate the true prevalence of EDS/IN. At the primary care level, these symptoms are defined as (1) very likely or certainly falling asleep during daytime in passive situations (reading/watching TV, at the cinema, during a lecture, as a passenger on public transport, resting in the afternoon) or in active situations (at work, while driving, during conversations/meetings); and (2) subjective sleep disturbances such as difficulty initiating sleep, difficulty maintaining sleep, and waking up earlier than desired or non-restorative sleep.

Every PCP will use the same primary care data collection sheet and send it anonymized to the study team in Bern on a regular basis at least once a month. Potential participants who express interest in participating in the study will sign a separate short contact data form from a PCP in order to be contacted by the study center. Irrespective of whether subjects decide to participate in the study or not, individuals with EDS or IN symptoms will be asked if they would like to receive a diagnostic workup and therapy from sleep specialists.


*Screening*


Potential participants who agree to participate in the study will be contacted by the study center in Bern via telephone, during which the eligibility criteria listed in Table 1 will be checked. Subjects who meet these criteria and express their willingness to participate will be sent the Informed Consent Form (ICF) via postal mail from the study center. Subjects who sign the ICF will receive a link to the online survey with the ESS and ISI questionnaires via email. If the subject’s subjective excessive daytime sleepiness and insomnia are confirmed with ESS [13] and/or ISI [14] scores within the required range (ESS > 10, ISI > 7), the subject will be enrolled.

Based on the screening experience in Bern before the trial becomes multicenter, it is possible that the eligibility criteria and recruitment process could be adapted to better suit local requirements and primary care level facilities.

Baseline and Follow-Up

All the data will be collected directly from participants, using online questionnaires, sent individually to each participant via email. At baseline, participants will complete the most time-consuming survey, necessitating approximately 20–25 min. The follow-up surveys are less time-consuming and are meant first of all to check for any changes in status, if applicable.

To assess the burden of sleep/wake disorders among participants with EDS/IN complaints, we aim to consider multiple dimensions such as health-related quality of life, health aspects such as BMI, incidence of accidents, and productivity, as well as economic aspects. For this purpose, we will apply standardized questionnaires (SF-12 [15], EQ-5D-5L [16], ESS, ISI, PSQI [17]) and additional questionnaires specifically designed for this study based on the previous experience of studies with assessment of socio-economic burden of different neurological diseases. With this approach, we hope to obtain comprehensive data to describe the burden of sleep/wake disorders from different perspectives.

Participants are planned to be followed for up to 12 months with the expectation of observing significant changes in sleep/wake complaints and associated burden of sleep/wake disorders. Due to feasibility constraints, the frequency of follow-up surveys was decreased to two (one at baseline and one after 12 months). To evaluate the dropout rates and potential benefits for participant retention, the final 20 participants are additionally administered a reduced set of standardized questionnaires every three months.

At the baseline, all the participants will be asked if they are interested in receiving a diagnostic workup and possible treatment for EDS or IN. However, this is not part of the eligibility criteria as we do not aim to assess the effectiveness of treatment in this study. The treatment administered during the study will be monitored through follow-up questionnaires. Participants who consent will be referred to their primary care physician for an appointment at a specialized medical institution. For those who decline further evaluation for EDS or IN, the reasons for their decision (e.g., no perceived illness, mild severity, unresponsive to therapy, lack of time, cost concerns, other reasons) will be recorded.

Figure 1 shows the study flow chart. Primary care level patients with subjective daytime sleepiness and insomnia symptoms assessed with ESS and ISI complete an online survey including standardized quality of life and sleep quality questionnaires, as well as specially designed health economic questionnaires with follow-up until 12 months.

Assessment of outcomes

*1*.
*Prevalence of EDS/IN in the primary care setting.*


The prevalence of EDS/IN complaints will be assessed after the completion of prescreening by primary care physicians using a primary care data collection sheet. We anticipate pre-screening of approximately 600 patients consecutively to reach the number of 100–120 potential participants. A basic demographic description of this subset of the population will be performed, recording age, sex, duration of symptoms, and current treatment. In addition, the level of concern of the patients about the complaints will be determined. At the primary care level, a streamlined methodology will be utilized that involves direct inquiry into the patient’s sleep/wake complaints. The subjective assessments, including the likelihood of falling asleep during passive or active situations, difficulties with sleep onset and maintenance, and non-refreshing sleep, will be then measured using the ESS and the ISI questionnaires during screening procedures.

*2*.
*Association between EDS/IN and QOL and BMI.*


Subjective complaints will be assessed with ESS and ISI questionnaires as part of screening procedures.

Assessment of excessive daytime sleepiness and insomnia

The Epworth Sleepiness Scale (ESS) is a self-administered questionnaire used to assess a person’s daytime sleepiness. The ESS consists of eight questions asking the likelihood of dozing off in various daily activities. The score ranges from 0 to 24, with higher scores indicating greater levels of sleepiness, a cut-off value of 10 is used in this study.

The Insomnia Severity Index (ISI) is a 7-item self-report questionnaire used to evaluate the severity of insomnia symptoms. The ISI assesses the impact of insomnia on daily functioning, the degree of distress caused by insomnia, and sleep quality. The scores range from 0 to 28, with higher scores indicating more severe insomnia, a cut-off value of 7 is used in this study. The ISI is widely used in clinical and research settings as a quick and efficient way to evaluate insomnia symptoms.

Assessment of quality of life

The SF-12 is a shortened version of the SF-36 Health Survey, a commonly used tool to assess health-related QOL. The SF-12 measures both physical and mental health using 12 questions, covering domains such as physical functioning, role limitations due to physical health, and emotional well-being. The scores are transformed into a 0–100 scale, with higher scores indicating better health-related quality of life. The SF-12 has been validated in a variety of populations and has been shown to be a reliable and valid measure of health-related quality of life.

The EQ-5D-5L is a standardized questionnaire used to assess health-related quality of life. It consists of five dimensions: mobility, self-care, usual activities, pain/discomfort, and anxiety/depression. The questionnaire asks the respondent to rate their level of problems in each dimension on a five-level scale, ranging from “no problems” to “extreme problems”. The EQ-5D-5L is used in both clinical and research settings and is widely used in cost-effectiveness studies to assess the impact of health interventions on quality of life.

Body Mass Index (BMI)

All participants complete baseline questionnaires containing questions about their current weight and height. Based on the self-reported data, the BMI is calculated by dividing an individual’s weight in kilograms by the square of their height in meters.

*3*.
*Association between EDS/IN and:*



*Presence of comorbidities and medication use;*

*Number of injuries/accidents (with 1–3 days sick days or with >3 sick days) in the 3 months prior to baseline;*

*Number of sick/leave days in the 3 months prior to baseline for the subgroup of working study participants.*


Online questionnaires will be used at baseline to collect information on comorbidities, medication use, retrospective data on injuries or accidents in the past three months, and the number of sick/work days lost in the past three months.

*4*.
*Change from baseline after 12 months*


After 12 months all the participants will be invited to complete online questionnaires (ESS, ISI, SF-12, EQ-5D-5L) and to collect information about the following:Current weight;Medication use;Number of injuries/accidents (with 1–3 days sick days or with >3 sick days) in the 3 months prior to 12 months follow-up;Number of sick/leave days in the 3 months prior to 12 months follow-up for the subgroup of working study participants.

These data will be compared to baseline to track changes in excessive daytime sleepiness and insomnia and associated changes in quality of life, BMI, medication use, number of injuries and accidents, and sick/leave days for the subgroup of working study participants. The summary of assessments is presented in the Table 2.

Data management and study monitoring

The CRFs in this trial will be implemented electronically using a dedicated EDC system (REDCap, https://www.project-redcap.org/, accessed on 28 August 2024). The EDC system will be activated for the study only after successfully passing a formal test procedure. All data entered in the CRFs will be stored on a Linux server in a dedicated mySQL database. Responsibility for hosting the EDC system and the database lies with the Clinical Trial Unit CTU of the University of Bern. Once informed consent is obtained and the participant is registered in the EDC system, an automatic email with the link to the questionnaire will be sent to the participant.

Study personnel, according to the staff list, will be trained on all aspects of the study as required for them to properly execute their assigned tasks. Data will be checked by the electronic data capturing (EDC) system for completeness and plausibility using built-in automatic validation checks. Furthermore, selected data points will be cross-checked for plausibility with previously entered data.

Statistical analysis

The analysis will be exploratory and will primarily make use of descriptive statistical methods. Inferential statistical methods will be used to highlight interesting aspects of the data. Statistical tests will be conducted at the 5% significance level unless otherwise specified, and corresponding 95%-confidence intervals for parameter estimates will be computed as appropriate. As no confirmatory analysis is planned, no formal correction for multiple testing will be applied.

The primary analysis population will include all recruited subjects (full analysis set). For specific questions, subjects not fulfilling the eligibility criteria may be excluded.

The primary analysis will be performed after the last patient has left the study.


*Sample size*


For the 10 participating PCPs collectively, we assume a realistic total recruitment potential of 100 patients over a recruitment period lasting until the end of June 2024.

The target sample for this exploratory study is primarily based on feasibility considerations but is considered adequate in relation to the objectives of the study. Assuming 20% of non-evaluable subjects, the study would have 80% power to detect a linear correlation (e.g., between the ISI and the SF-12 score) event with a coefficient as low as 0.17. Furthermore, the precision (half-width of the 95% CI) of estimates of binary endpoints, such as the prevalence of a given comorbidity, will be <6.5% (for a 50% prevalence) and will be 4% for a 10% prevalence.

Safety

If, during this study, circumstances arise that could jeopardize the safety or health of the participants or lead to a disproportionate relationship between the risks and burdens and the benefits, all the measures required to ensure protection will be taken without delay. The local project leader and the sponsor will be promptly notified (within 24 h) if immediate safety and protective measures have to be taken during the conduct of the study.

Discussion

“The Burden of Sleep/Wake Disorders” is a retro- and prospective observational study with a focus on the main sleep complaints such as excessive daytime sleepiness and insomnia, which can be key factors in decreased quality of life and suffering from a patient’s perspective. Due to the evidence of a higher rate of sleep/wake disorders such as chronic insomnia and sleep apnea at the primary care level, the population subset sample is chosen from the PCP’s routine patients. The pilot study will provide information on the feasibility, prevalence of sleep/wake complaints in the primary care setting, and association between EDS/IN and health and quality of life, as well as socio-economic burden. The present state-of-the-art can highlight the issue of underdiagnosis of sleep/wake disorders and the importance of addressing this issue through policy improvement. This includes prioritizing the collaboration between primary care physicians and specialists to bridge the gap in providing adequate therapy for patients. The underdiagnosis of sleep/wake disorders is a significant challenge and its burden is often underestimated, therefore, concerted efforts are necessary to ensure its proper recognition and treatment.

Limitations of this study include the following: (1) The lack of a control group in a study impedes the evaluation of the effect of Excessive Daytime Sleepiness/Insomnia (EDS/IN) on health outcomes and health-related quality of life; (2) the presence of comorbid conditions can influence the results of a study, and their impact cannot be accurately assessed; and (3) the sample size of 100 participants in the pilot study raises uncertainty regarding its ability to detect significant changes after 12 months of follow-up; nevertheless, it is deemed sufficient for the assessment of the feasibility of the project, with the potential for future expansion into a multi-center study.

Trial status

Recruitment started in MM 2023 in Switzerland in collaboration with primary care physicians coordinated by the Bern University Hospital. Recruitment is expected to be completed in June 2024. The current version of the protocol is version 1.1 dated 15 May 2023.

## Figures and Tables

**Figure 1 mps-07-00070-f001:**
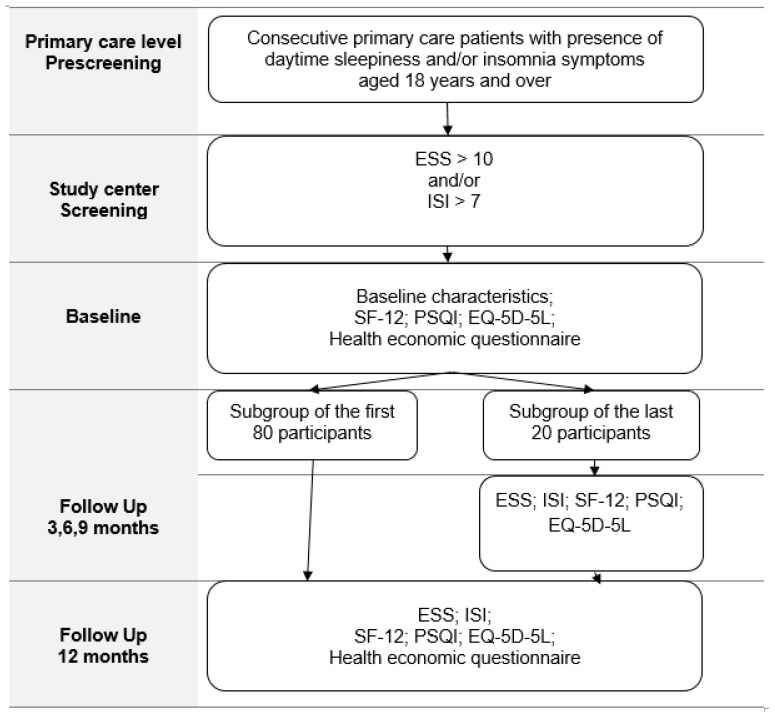
An overview of the study design. Abbreviations: ESS, Epworth Sleepiness Scale; ISI, Insomnia Severity Index; SF-12, Short Form survey with 12 items; PSQI, Pittsburg Sleep Quality Index; EQ-5D-5L, The 5-level health-related quality of life questionnaire from EuroQOL Group.

**Table 1 mps-07-00070-t001:** Eligibility criteria.

Per Telephone/Post	Online Survey
Informed consent as documented by signature Age ≥18 years Ability to speak and understand German or French Ability to use an electronic device to complete questionnaires online Subject does not plan to leave Switzerland within the next 12 months starting from screening/baseline	Epworth Sleepiness Scale (ESS) > 10 and/or Insomnia Severity Index (ISI) > 7

**Table 2 mps-07-00070-t002:** Summary of assessments at each time point.

Visit and Location	1 Pre-Screening	2 Screening and Baseline		3, 4, 5 ^1)^ Follow-Up	6 Follow-Up
Location	GP Practice and Pharmacy	Telephone and Online	Online	Online	Online
Time Point	Up to −26 Weeks	Up to −14 days	0	+3,6,9 Months	+12 Months
Pre-screening using “Primary care data collection sheet”	+				
Oral and written information and consent		+			
ESS and ISI			+	+	+
Medical history and treatment plan			+		+
Participant characteristics			+		
Sleep-related questions			+		
PSQI, SF-12			+	+	+
Health economic questionnaire, EQ-5L-5D			+		+
BMI			+		+

^1)^ Follow-up visits 3, 4, and 5 will only be performed with the last 20 participants enrolled. At months 3, 6, and 9, participants will be invited to complete the questionnaires online but will not be reminded to do so via telephone.

## Data Availability

No data were analyzed for the manuscript.

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
