# Peer review of "The Burden of Sleep/Wake Disorders: Excessive Daytime Sleepiness and Insomnia Project"

_mps, 2024, doi:10.3390/mps7050070_

Round 1

Reviewer 1 Report

Comments and Suggestions for Authors

Dear authors,

I have read with interest the paper entitled “The Burden of Sleep/Wake disorders: Excessive daytime sleepiness and insomnia project”. In this paper, the authors present the design of a retro and prospective monocentric cohort observational study for the systematic evaluation of the burden of excessive daytime sleepiness (EDS) and insomnia. EDS and insomnia will be evaluated through validated scales and patients with pathological cut-offs and who agree to participate will have to complete further scales assessing health and socio-economic burden at the baseline and after 12 months. This pilot study means at evaluating the feasibility of such a study, in order to conduct a European multicenter study on the burden of sleep/wake disorders.

The paper is well written and its aim is clear. I have some minor concerns which should be addressed by the authors.

The authors state at lines 198-201 that “at the baseline all the participants are asked about interest to receive a diagnostic workup and possible treatment for EDS or IN. However, this is not part of the eligibility criteria as we don’t aim to assess the effectiveness of treatment in this study”. If some patients decide to receive treatments for their complaints and others not, will this variable be taken into account at 12-monts follow-up? This aspect is not clear in the text and the authors should specify it.

Try to avoid repetitions between the aims of the study and the study design paragraph. 

Comments on the Quality of English Language

Minor: check spelling line 47 “und”, line 88 “heir”, line 144 “:”.

Author Response

Dear Reviewer,

Thank you for your thorough review and insightful comments on our manuscript. We appreciate the time and effort you have dedicated to providing constructive feedback, which has significantly contributed to improving the quality of our work. In response to your suggestions, we have made the following revisions and additions to the manuscript:

Comment 1: The authors state at lines 198-201 that “at the baseline all the participants are asked about interest to receive a diagnostic workup and possible treatment for EDS or IN. However, this is not part of the eligibility criteria as we don’t aim to assess the effectiveness of treatment in this study”. If some patients decide to receive treatments for their complaints and others not, will this variable be taken into account at 12-monts follow-up? This aspect is not clear in the text and the authors should specify it.

Reply 1: Addition: Lines 201-202 now include the statement: "The treatment administered during the study will be monitored through follow-up questionnaires."

Comment 2: Try to avoid repetitions between the aims of the study and the study design paragraph. 

Reply 2: lines 129-131 deleted. "The pilot study is conducted to assess the feasibility and efficacy of a larger, multicenter study at the European level, evaluating the involvement of primary care physicians in the recruitment process, implementation of online questionnaires, and follow-up for a period of 12 months."

Comment 3: Minor: check spelling line 47 “und”, line 88 “heir”, line 144 “:”.

Reply 3: All corrected to "and", "their" and ":" deleted. 

Thank you for your consideration.

Best regards,

Marina Tüzün

Reviewer 2 Report

Comments and Suggestions for Authors

This protocol/study design manuscript is clear and concise, and provides a good overview of a pilot trial to inform a European-wide study to better understand the prevalence of broad sleep problems, namely excessive daytime sleepiness and insomnia. The study design is simplistic but there is value in its simplicity to assess prevalence of sleep problems that can then inform potential practice/policy changes to improve patient care and care delivery. In general, some minor issues to consider that can improve the manuscript. 

Introduction

Line 65-66: When reporting on prevalence of sleep disorders in Europe (45 million), provide % as what is listed is likely mostly sleep apnea and insomnia with narcolepsy and hypersomnia much less common. 

Line 88-89: Can further comment that there are actually dozens of measures on sleep disorders but there are also few standards and consensus measures, and many are behind a paywall and not freely available for research and/or clinical use. This can prevent common measures between studies. 

Line 101-103: How will the benefits of diagnosing and treating EDS/INS be measured/operationally defined?

Experimental Design

Line 131-132: Can delete as same text in next paragraph.

Line 145: Any telehealth visits or only in-person/in-office visits? Also, any questions related to chronicity of symptoms?

Line 197: Interesting approach to test feasibility/burden of more frequent testing. Although online questionnaires/REDCap are low effort and it could be worthwhile to try and collect data from all participants. This would also provide more feasibility data of doing more frequent follow-up at a larger scale. 

Line 201: Will you track who accepts the referral/consult and who engages (or not) in treatment? This is incredibly valuable data, even if not tracking effectiveness as engagement data in sleep-related treatments is lacking, both initiation, completion, drop out, etc. Is this data able to be tracked/collected in an electronic medical record (is there a standard system used in Switzerland)? Agree with decision to collect data on those who choose decline further evaluation. 

Table 2: Are you collecting any variables related to social determinants of health and health equity. Variables such as distance from a clinic/provider, access to transportation, access to technology/telehealth, technology understanding/awareness, etc. 

For the 20 participants with 3, 6, 9 month follow-ups, there is no telephone reminders. Does this mean that for baseline and 12 month assessments, there is a telephone reminder, and does this occur before REDCap link is sent or only if no response in a certain time frame?

Discussion

Line: 340-342: "The under diagnosis of sleep/wake disorders is a significant challenge and its burden is often underestimated, therefore, concerted efforts are necessary to ensure its proper recognition and treatment." Can you expand on identified and/or proposed barriers and facilitators? What policy changes may be needed to improve clinical efforts for this population? Any hypotheses or proposed changes at this stage of research to test or confirm?

Line 343: Why not collect a control group of patients who screen negative for the two sleep questions? To consider in a larger study. Can help with evaluating the development of sleep problems and potential transition from acute to chronic problems and potential identification of risk factors for new onset EDS/INS.

Author Response

Dear Reviewers,

Thank you for your thorough review and insightful comments on our manuscript. We appreciate the time and effort you have dedicated to providing constructive feedback, which has significantly contributed to improving the quality of our work. In response to your suggestions, we have made the following revisions and additions to the manuscript:

Comment 1: Line 65-66: When reporting on prevalence of sleep disorders in Europe (45 million), provide % as what is listed is likely mostly sleep apnea and insomnia with narcolepsy and hypersomnia much less common. 

Reply 1: In response to your feedback, I have revised the text to include the percentage information from the same source: "including sleep apnea affecting 3% of the population, chronic insomnia affecting up to 7%, narcolepsy at 0.022%, and hypersomnia at 0.75%."

Comment 2: Line 88-89: Can further comment that there are actually dozens of measures on sleep disorders but there are also few standards and consensus measures, and many are behind a paywall and not freely available for research and/or clinical use. This can prevent common measures between studies. 

Reply 2: Thank you for the addition, we agree to the comment. This is a significant challenge that warrants further discussion in the field.

Comment 3: Line 101-103: How will the benefits of diagnosing and treating EDS/INS be measured/operationally defined?

Reply 3: Added to the sentence: ", along with the potential benefits of their diagnosis and treatment—such as enhancements in quality of life, sleep quality, and daytime functioning—remains to be explored."

Comment 4: Line 131-132: Can delete as same text in next paragraph.

Reply 4: deleted

Comment 5: Line 145: Any telehealth visits or only in-person/in-office visits? Also, any questions related to chronicity of symptoms?

Reply 5: For the pilot study, only in-person visits at the primary care office will be conducted. During these visits, data will be collected using the primary care data collection sheet, including information on whether symptoms have persisted for more than 3 months, 1 to 3 months, or less than 1 month.

Comment 6: Line 197: Interesting approach to test feasibility/burden of more frequent testing. Although online questionnaires/REDCap are low effort and it could be worthwhile to try and collect data from all participants. This would also provide more feasibility data of doing more frequent follow-up at a larger scale. 

Reply 6: Thank you for your feedback.

Comment 7: Line 201: Will you track who accepts the referral/consult and who engages (or not) in treatment? This is incredibly valuable data, even if not tracking effectiveness as engagement data in sleep-related treatments is lacking, both initiation, completion, drop out, etc. Is this data able to be tracked/collected in an electronic medical record (is there a standard system used in Switzerland)? Agree with decision to collect data on those who choose decline further evaluation. 

Reply 7: Unfortunately, we can only collect data directly provided by participants and cannot access additional documentation due to strict data protection regulations and ethical guidelines in Switzerland. Additionally, there is no central data collection system available for studies.

Comment 8: Table 2: Are you collecting any variables related to social determinants of health and health equity. Variables such as distance from a clinic/provider, access to transportation, access to technology/telehealth, technology understanding/awareness, etc. 

Reply 8: We do not plan to include this in the pilot study; however, it could be considered for the multi-center study, we will refine the protocol based on the findings from the pilot. Thank you for raising this important point.

Comment 9: For the 20 participants with 3, 6, 9 month follow-ups, there is no telephone reminders. Does this mean that for baseline and 12 month assessments, there is a telephone reminder, and does this occur before REDCap link is sent or only if no response in a certain time frame?

Reply 9: RedCap automatically sends reminders with links, but to maximize information collection, we need to maintain phone contact in case these reminders go unanswered. I don’t believe we can rely solely on automated reminders for this purpose.

Comment 10: Line: 340-342: "The under diagnosis of sleep/wake disorders is a significant challenge and its burden is often underestimated, therefore, concerted efforts are necessary to ensure its proper recognition and treatment." Can you expand on identified and/or proposed barriers and facilitators? What policy changes may be needed to improve clinical efforts for this population? Any hypotheses or proposed changes at this stage of research to test or confirm?

Reply 10: One potential focus for further discussion could be enhancing primary care providers' awareness of the prevalence of sleepiness and insomnia symptoms within their patient population. Additionally, evaluating the feasibility of pre-screening tools used in the primary care sheet could be beneficial. Although these aspects may not have been specifically addressed in the pilot study, they can be set as goals for the next working group to explore after the initial steps are completed.

Comment 11: Line 343: Why not collect a control group of patients who screen negative for the two sleep questions? To consider in a larger study. Can help with evaluating the development of sleep problems and potential transition from acute to chronic problems and potential identification of risk factors for new onset EDS/INS.

Reply 11: Indeed, including a control group is an important consideration for the multi-center study following the pilot. Due to resource constraints and feasibility issues, we were unable to include a control group in Switzerland, unfortunately.

Thank you for your support and consideration!

Best regards,

Marina Tüzün